# Is There an Association between Viral Infections and Risk for Pediatric Obstructive Sleep Apnea? A Systematic Review

**DOI:** 10.3390/children10030487

**Published:** 2023-03-02

**Authors:** Elody Aïem, Clémence Leblais, Laurence Lupi, Alain Doglio

**Affiliations:** 1Faculté de Chirurgie Dentaire Odontologie, Université Côte d’Azur, 5 rue du 22e B.C.A, 06000 Nice, France; 2Institut de Médecine Bucco-Dentaire, CHU Nice, 28 Boulevard de Riquier, 06300 Nice, France; 3Laboratoire Microbiologie Orale, Immunothérapie et Santé (MICORALIS) UPR7354, Campus Saint Jean d’Angely-SJ2, Faculté de Chirurgie Dentaire, 5 rue du 22e B.C.A, 06300 Nice, France

**Keywords:** obstructive sleep apnea, pediatric, viral infections, association, systematic review

## Abstract

(1) Background: Obstructive sleep apnea (OSA) affects approximately 1% to 5% of children. To date, the main pathophysiological factor is adenotonsillar tissue hypertrophy. As many respiratory viruses can persist in secondary lymphoepithelial organs after upper airway infection, the objective of this systematic review was to investigate the link between history of viral infections and the risk of pediatric OSA. (2) Methods: Corresponding references were searched electronically (PubMed [MEDLINE], Cochrane Library and Scopus) until 21 November 2022. Prospective or retrospective cohorts, evaluating the children suffering from OSA with history of viral infections and comparing them with children with no history of viral infections written in English, were included. Four independent reviewers selected studies, extracted data, and evaluated the risk of bias using ROBINS-I. (3) Results: Of 1027 potentially eligible articles, four studies (one retrospective, two prospective cohorts and one case-control) were included. (4) Conclusions: Exposure to lower airway infections may precede the diagnosis of pediatric OSA suggesting that respiratory viruses may play a mechanical role in the development of pediatric OSA. Further research is required to improve our understanding of the role of viral infections. Registration: PROSPERO CRD awaiting.

## 1. Introduction

Obstructive sleep apnea (OSA) is a common and widespread form of sleep-disordered breathing in children affecting from 1% to 5% aged 2–8 years [1,2]. It is characterized by repeated partial or full obstruction of the upper respiratory tract during sleep with episodes of apnea and hypopnea. Cardiovascular, neurocognitive and behavioral morbidities are associated with OSA [3,4,5]. Indeed, OSA enhances the risk of impaired cognitive function [6], attention deficit hyperactivity disorder [7], poor quality of life [8], enuresis [9], metabolic syndrome and cardiovascular disease [10]. For this reason, early diagnosis and adequate care are required to prevent long-term complications.

Palatine tonsils, as secondary lymphoepithelial organs, are fundamental for immune responses to respiratory pathogens and allergic antigens. It is also known that some viruses, such as Epstein–Barr virus (EBV), human herpes virus 6, human immunodeficiency virus, measles virus and enterovirus (EV), can contaminate the adenotonsillar tissues (AT) after an upper airway infection and may lead to persistent infection in an asymptomatic form during asymptomatic states [11,12,13,14,15]. Viral infections are prone to causing an inflammatory tonsil response leading to AT hypertrophy, which is by far the most important pathophysiological factor of OSA in children [16,17]. Right now, tonsillectomy and adenoidectomy remain the most effective and proposed treatment for children’s OSA [18]. Nonetheless, the pathogenesis of AT hypertrophy leading to OSA in early childhood is poorly understood.

In this regard, we speculated that children with a history of viral infections would be more likely to have OSA. Nevertheless, as far as we know, there has been no systematic review (SR) of this topic.

## 2. Methods

### 2.1. Protocol Registration

A SR was performed according to the Preferred Reporting Items for Systematic Reviews and Meta-Analyses (PRISMA) 2020 guidelines [19]. The protocol was incorporated in the International Prospective Register of Systematic Review (PROSPERO) (CRD awaiting).

### 2.2. Research Question and Eligibility Criteria

The research question was established as follows: “Is there an association between viral infections and risk for pediatric OSA?” 

Regarding the PECO framework:-P (Population): children with history of viral infection.-E (Exposition): viruses implicated in viral infections, such as respiratory infections.-C (Comparison): children with absence of viral infection.-O (Outcome): development or diagnosis of OSA.

In order to be included in the SR, the studies had to fulfill the following criteria: prospective or retrospective, evaluating the association between viral infections and development of OSA in children and written in English without limitation in the publication date. The exclusion criteria were as follows: randomized controlled trials (RCTs) or nonrandomized trials (controlled clinical trials), non-comparative studies (case reports and case series), editorials, opinions, reviews, in vitro studies, experimental studies, SR or meta-analyses.

### 2.3. Information Sources and Research Approach

A comprehensive literature research was conducted using PubMed (MEDLINE), Cochrane Library and Scopus in order to determine which studies were eligible on 21 November 2022 (Table 1). In addition, a manual search was carried out in the bibliography for each of the articles included to discover articles that had not been found through the electronic search.

### 2.4. Study Selection

At least two authors independently screened the titles and abstracts of the publications identified in the electronic search and then evaluated the entire report for each potentially relevant study. The disagreements were settled by discussion with a third reviewer.

### 2.5. Data Collection Process

Two reviewers independently extracted the data using a piloted data extraction form, and any disagreements were resolved by discussion: first author’s name, year of publication, country, study design, inclusion and exclusion criteria for subjects, protocol, comparison and results. Pooling results into a meta-analysis was not feasible. As a result, the analysis was qualitative and descriptive.

### 2.6. Assessment of Risk of Bias of the Included Studies

Two authors independently assessed the risk of bias according to the Cochrane Handbook for Systematic Reviews of Interventions Version 6.3 using the ROBINS-I (Risk of Bias in Non-randomized Studies-of Interventions) tool [20]. They compared the assessments and solved disagreements through discussion. When required, a third author was consulted. The Risk of bias figure was realized using the robvis tool [21].

## 3. Results

### 3.1. Study Selection

The research resulted in 1027 potential eligible records. Seven were selected for the integral text pre-selection and four studies were included [22,23,24,25] (Figure 1). 

### 3.2. Study Characteristics

The features of each study are set out in Table 2 and Table 3. Among the four studies, one was a case-control study [24], one was a retrospective cohort [22] and two were prospective cohorts [23,25] released between 2009 and 2021. They were conducted in the United States of America [22,23], Israel [24] and Taiwan [25]. Two studies were published in *Pediatric Pulmonology* [22,24], one in *Pediatric Infectious Disease Journal* [26] and one in *Sleep*, which is the official journal of the Sleep Research Society [23]. 

For both prospective cohorts, the mean durations of follow-up period ranged from 5 [23] to 9.67 and 9.72 years [25]. 

In four studies, the cofounding variables taken into account in statistical analysis were different. In two studies, all patients were Caucasian [24,25], whereas in the two other studies [22,23] there was a racial mixing, but the majority were black. 

In three studies [22,23,24], obesity was taken into consideration at baseline, but only BMI categories (18.5–≥30—normal, underweight, overweight and obese) were statically significant [23] while BMI Z score was not [22,24]. 

Among the two studies [22,24] evaluating tonsil size using Brodsky scores [27], only one revealed a significant difference [22].

The diagnosis of OSA was strictly based on electronic medical records (EMR) depending on International Classification of Diseases (ICD-9/ICD-10) codes [23,25] or a formal diagnostic process [22,24]. Indeed, the analysis of the polysomnograms was performed in one study [24] with the main criteria posted by the American Academy of Sleep Medicine in 2007 and in the other [22] with the American Sleep Disorders Association Task Force statement [22,28,29]. Both studies showed significant oxygen saturation as measured by pulse oximetry and the apnea hypopnea index (AHI). However, in only one study polysomnograms were blindly examined and independently interpreted by two doctors with experience in pediatric polysomnography [24]. 

With regard to the viruses, seven (influenza B, influenza A, human metapneumovirus (hMPV), adenovirus, parainfluenza 3 virus, rhinovirus and coronavirus NL63) were investigated in one study [24], two studies only investigated EV [24,25] and three studies investigated respiratory syncytial virus (RSV) [22,23,24].

### 3.3. Risk of Bias in Included Studies

Of the four studies, three were reported to be at serious risks of bias [22,24,25] and one [23] was reported to be at moderate risk of bias (Figure 2). In essence, the highest risk of bias is related to the performance measurement of outcomes (D6) (Figure 2).

### 3.4. Results of Studies

For all four studies, important differences were found between the control and the OSA groups. All findings are provided in Table 3.

## 4. Discussion

The present SR showed OSA is mostly occurring among children with a previously diagnosed lower respiratory tract infection (LRTI) [23], EV infection [25] or RSV infection [22,23,24] during early childhood.

However, nine viruses (RSV, influenza A, influenza B, hMPV, adenovirus, parainfluenza 3 virus, rhinovirus, coronavirus NL63 and EV) have been identified in this SR, the majority of which were responsible for acute respiratory viral infection in children. However, this does not preclude the potential role of other viruses, such as certain viruses from the herpes virus that can promote long-term persistent viral infection in the palatine tonsils.

In an overnight sleep study, Snow et al. first described children whose RSV bronchiolitis in early childhood had an obstructive apnea/hypopnea index and a significantly larger tonsil size than controls. Consequently, a relatively low number of children (21 with OSA versus 63 controls) were involved, and these subjects account for just a minor fraction of all children admitted to hospital during babyhood [22]. Furthermore, within the control group, children were defined from multiple databases obtained from anterior studies. However, the authors provide no information on the prior studies used.

In the case-control study, Yeshuroon-Kuffler et al. found that all (34) children in the OSA group had a history of a LRTI more than one year before the procedure, and nucleic acids of respiratory pathogens (such as rhinovirus, adenovirus and hMPV) were significantly more recurrent in their palatine tonsils. In the same way, Proenca et al. examined tonsils that were withdrawn due to diverse indications and indicated adenovirus, rhinovirus and hMPV to be the most frequent viruses in children with adenotonsillar diseases. Nevertheless, the diagnosis of OSA (only 11 children or 9.1%) was uncertain because no details about the method of diagnosis for OSA were given [31]. 

In a large birth cohort, Gutierrez et al. established that the adjusted hazard ratio (HR) for the occurrence of OSA during the first 5 years after early-life LRTI was 1.53 (95% CI 1.15 to 2.05, *p* = 0.004) regardless of mother characteristics (race/ethnic origin, BMI category before pregancy, mother’s educational level and smoking during pregnancy), children’s baseline characteristics (sex and preterm category), breastfeeding and preschool overweight [23]. In addition, infants with severe RSV bronchiolitis have an increased risk of OSA in the early years of their life (OR = 2.09, 95% CI 1.12 to 3.88, *p* = 0.020) [23].

Golbart et al. found significantly superior expression rates of nerve growth factor (NGF) mRNA and its high affinity tyrosine kinase receptor (trkA) in AT harvested from 34 children with OSA when compared to AT collected from 25 children with recurrent tonsillitis. Additionally, neurokinin 1 (NK1) receptor mRNA and protein expression and protein levels of substance P also increased significantly in the OSA group (*p* < 0.001) [32]. Hence, the authors supposed that early RSV infection might lead to long-term alterations in the neuro-immunomodulatory pathways of adenotonsillar tissue and may predispose such tissues to accelerated proliferative responses when exhibited to stimuli, such as viruses, allergens, etc., that are related with an increased prevalence of OSA in pediatric populations [32]. Neurogenic inflammation (NI) is defined as an inflammatory process that is activated and perpetuated by neurons near the airways as a result of viral infection [33]. Indeed, NI is recognized as one of the major processes linking exposure to viral infections, such as RSV in early childhood, to chronic inflammatory diseases, such as early asthma [34].

Faden et al. noted that EBV in palatine tonsils was linked to a higher degree of airway obstruction appraised by the Brodsky scores [27] (high scores of 3 or 4) (*p* > 0.03) [26].

The link between viral infections and OSA is also highlighted by a nationwide population-based cohort study in Taiwan that found a notable relationship between EV infection with hospitalization and pediatric OSA after adjusting for age, sex, urbanization, perinatal complications and atopic disease [25]. However, the diagnosis of sleep disorders was entirely based on diagnostic codes in the format of the International Classification of Disease, Revision 9, Clinical Modification (ICD-9-CM) rather than a formal diagnosis process. In fact, OSA diagnosis is based on PSG, which is the gold standard [35].

Gutierrez et al., in an important prospective Boston Birth Cohort (*n* = 3 114), identified that LRTI taken place during the first two years of life significantly improved the risk of pediatric OSA in pre-school years independently of major risk factors for OSA and other pediatric respiratory diseases (e.g., prematurity or obesity) [23]. However, the diagnosis of OSA relied solely on the EMR using an ICD-9/ICD-10 algorithm, and the LRTI definition did not contain confirmation of the virus. Additionally, some determinants (e.g., familial history of atopic disease or OSA, relevant environmental exposure, nutritional and lifestyle factors and pertinent socio-economic information) were considered. Since the cohort was composed of a downtown minority population, the results could not be extrapolated to other populations. The authors imply that within the first 2 years of life, respiratory infections may alter the nasopharyngeal tract and make a critical contribution to the pathophysiology of pediatric OSA [23].

In an epidemiological study of childhood obesity in China including 203 overweight preschool children, numerous risk factors significantly correlated with OSA, in particular adenotonsillar hypertrophy (as defined by a swelling of the tonsils on the arches of the pharynx) (odds ratio [OR] 3.52, *p* < 0.01) and recurrent respiratory tract infection (OR 2.57, *p* < 0.01) [36]. 

Thus, it is possibly the immune response to the virus which allows the virus to remain only in the OSA group. Tan et al. discussed how pediatric OSA is correlated with a decrease in the regulatory T lymphocyte population and a shift from Th1:Th2 equilibrium to Th1 predominance. This disorder is not only proinflammatory, but also associated with an altered response of T lymphocytes to some stimuli [37]. It is critically important to understand the unique nature of respiratory organs’ mucosal immune systems (such as tonsils) for future prevention and treatment of OSA.

Nevertheless, the fact that the exposure to LRTI lead to the diagnosis of OSA indicates that respiratory viruses may contribute mechanically to the development of pediatric OSA and more investigation is warranted to determine if early-life LRTI could be a cause and/or a marker for an subjacent vulnerability to OSA in children [23].

### Strength and Limitations

This was the first SR on the potential effect of a viral infection on OSA in childhood that employed a systematic and objective method to identify as many relevant studies as possible in English to lend an extensive and impartial presentation of the topic.

We included prospective, retrospective cohorts and case-control studies, which examined the association between risk factors (viral infection) and the prevalence, experience or incidence of OSA. Case series, case reports and cross-sectional studies were not included. RCTs were also eliminated because an interventional study is not the best study design for assessing the association between the risk factor and disease onset. Our study was based on the PECO format.

As a whole, the included studies of OSA risk factors have three main limitations. The first is the failure to adjust for confounding factors. A known limitation of observational studies is the ability of confounders to overstate or diminish the significance of certain factors as randomization is not feasible. This is usually compensated by using multiple logistic regression analysis, which is almost compulsory in these studies. This analysis is dependent on the use of dichotomized data, which means that the categorizations used in each study can be as important as the number of exposures tested. For instance, one study [22] could investigate before the first birthday while another study could compare infections six months before surgery [24] and come to different conclusions. The second is the lack of coherence and detail between the risk factor categories studied, which limits comparison between studies. 

More uniformity among the studies to measure oral health outcomes (OSA) and the risk factors in children are required to facilitate a more precise knowledge base of the risk factors for OSA. The third limitation of the four included studies was that three were assessed to have a serious risk of bias [22,24,25] and one was assessed to have a moderate risk of bias [23]. In fact, only one study had a clear explanation of OSA diagnosis. These results are relevant to future research.

## 5. Conclusions

According to this SR, respiratory viruses can be implicated in developing AT hypertrophy and nasopharyngeal airway obstruction and pediatric OSA. Nonetheless, a presumed association between history of viral infections and pediatric OSA is poorly characterized. It is important to understand how viral infections interfere with respiratory tract homeostasis. To elucidate the early origins of pediatric OSA and prevent its multiple impacts on pediatric health and beyond, more studies are needed.

## Figures and Tables

**Figure 1 children-10-00487-f001:**
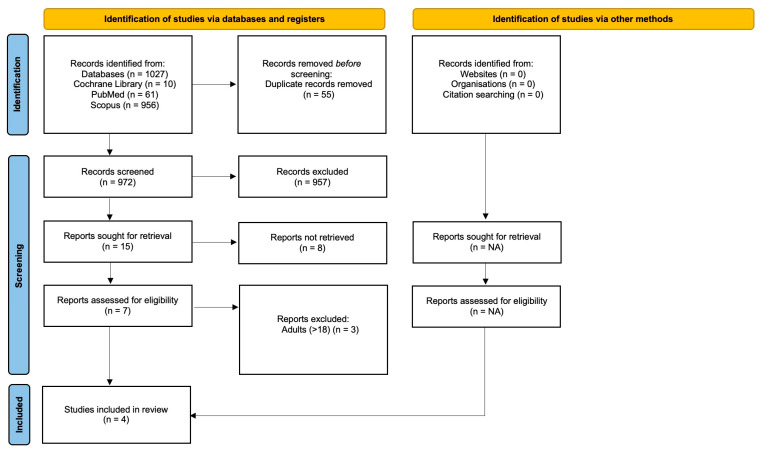
Preferred Reporting Items for Systematic Reviews and Meta-Analyses (PRISMA) flow diagram.

**Figure 2 children-10-00487-f002:**
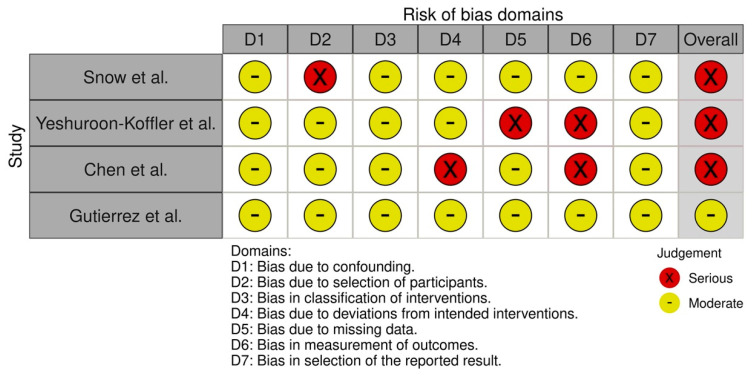
Risk of bias assessment in cohort and case-control studies according to the ROBINS-I assessment tool.

**Table 1 children-10-00487-t001:** Search strategy.

Database	Keywords
Cochrane Library	((child) OR (preschool)) AND ((Sleep Apnea, Obstructive) OR (sleep apnea syndromes)) AND ((viruses) OR (Diseases, Virus))
PubMeb (MEDLINE)	((child) OR (children)) OR (Child [MeSH Terms]) OR (Infants) OR (Infant [MeSH Terms]) OR (Adolescents) OR (Adolescent [MeSH Terms]) AND ((Sleep apnea, obstructive [MeSH Terms]) OR (Sleep apnea syndromes [MeSH Terms])) AND ((Viruses [MeSH Terms]) OR (Diseases, Virus [MeSH Terms]))
Scopus	(child OR children OR infant OR infants OR adolescent OR adolescents) AND (sleep apnea obstructive OR sleep apnea syndromes) AND (viruses OR diseases virus)

**Table 2 children-10-00487-t002:** Characteristics of included studies using PECO question. OSA = obstructive sleep apnea; AHI = apnea hypopnea index; BMI = body mass index; EV = enterovirus; hMPV = human metapneumovirus; LRTIs = lower respiratory tract infections; RI = recurrent throat infections; RSV = respiratory syncytial virus.

Authors	Journal and Year Published	Study Design	Population	Inclusion and Exclusion Criteria	Exposition	Comparison	Support
Snow et al. [22]	Pediatric Pulmonology, 2009	Retrospective cohort	84	- Exclusion criteria: past adenotonsillectomy, obesity (BMI z score > 1.67), presence of prematurity, admission to the pediatric intensive care unit, past intubation, chronic illnesses, genetic disorders, neuromuscular diseases, craniofacial abnormalities, presence of an additional acute infection during the bronchiolitis episode (as adenovirus or Influenzavirus A or B) or current treatment with medications, such as corticosteroids.	Hospitalized children for RSV infection before their first birthday.	Children without previously diagnosed RSV infection.	Yes
Yeshuroon-Koffler et al. [24]	Pediatric Pulmonology, 2015	Case-control	56	- OSA group: AHI > 5/h based on in laboratory overnight polysomnography. - Control group: Children without OSA diagnosis of RI, that is >5 Strept. Group A infections in the preceding 6 months prior to surgery. - Exclusion criteria: any previous history/physician diagnosis of cardiovascular disorder, allergies, smoking of a first degree relative/daytime caregiver (kindergarten teacher, grandmother), familial craniofacial or genetic disorders, upper airway infections 6 weeks prior to surgery and lower airway infection at least 12 months prior to surgery.	Diagnosis of RI, Strept. Group A infections 6 months prior to surgery.	Presence of viral genomic material.	Yes
Chen et al. [25]	Pediatric Infectious Disease Journal, 2018	Prospective cohort	95,718 * 96,012 **	- Children < 18 years of age before 2005 EV infection group included at least 2 ambulatory claims within 1 year or at least 1 inpatient claim for EV infection. - Exclusion criteria: sleep disorder diagnosis/sleep apnea diagnosis before EV infection.	EV infection.	Children without previously diagnosed EV infection.	No
Gutierrez et al. [23]	Sleep, 2021	Prospective cohort	3152	- Exclusion criteria: children with immunodeficiency, cystic fibrosis trisomy 21, or cleft palate and infants who had OSA during the first 3 months of life.	LRTIs during first 2 years of life.	Children without OSA.	Yes

* Only individuals with International Classification of Diseases (ICD) codes of nonapneic sleep disorder; ** only individuals with ICD codes of obstructive sleep apnea.

**Table 3 children-10-00487-t003:** Protocol and results of included studies. BBC = Boston Birth Cohort; BMI = body mass index; EMR = electronic medical record; EV = enterovirus; HAdV = human adenovirus; hMPV = human metapneumovirus; ICD-9-CM = International Classification of Diseases, Ninth revision, Clinical Modification; mqRTPCR = real time reverse transcription polymerase chain reaction; OSA = obstructive sleep apnea; OAHI = Obstructive apnea hypopnea index; RI = recurrent throat infections; RSV = respiratory syncytial virus; vs. = versus.

Authors	Protocol	Results
Snow et al. [22]	Screening charts from the medical files department at Kosair Children’s Hospital in Louisville, Kentucky. A total of 85/489 validated sleep questionnaire ^1^ posted to each family to be completed by one parent/guardian from each child were reviewed. A total of 43 eligible children were reached by telephone; 21 children suffered overnight polysomnography (Stellate, Montreal, Canada) and tonsil size was evaluated (and accorded a score from 0 to 4 *) at the Kosair Children’s Hospital pediatric sleep laboratory.	OAHI was significantly upper in the RSV group (2.3 ± 1.9 vs. 0.6 ± 0.8 events/h sleep; *p* < 0.05) as well as significantly higher respiratory arousal index (1.3 ± 1 vs. 0.1 ± 0.2 events/h sleep; *p* < 0.05). Tonsil size was raised in the RSV group (2.5 ± 0.9 vs. 1.2 ± 0.3; *p* < 0.01).
Yeshuroon-Koffler et al. [24]	Medical check (height, weight, and BMI). Examination of tonsil size * following a direct inspection of the oropharynx on the same doctor’s day of surgery. Overnight polysomnography conducted with a computerized, commercially available, sleep monitoring system (SensorMedics, Inc., Yorba Linda, California). Nucleic acids extraction: AllPrep DNA/RNA mini kit (Qiagen GmbH, Hilden, Germany), mqRTPCR and hydrolysis multiplex quantitative probes with internal control: human endogenous retrovirus ERV3 qRTPCR for beta-actin and RNAseP housekeeping genes. Panel 1: RTPCR System, Thermocycler 7500 (Applied Biosystems, FostervCity, CA); Panel 2: RTPCR System LC480 (Rhoche, Mannheim, Germany)	56 children: 34 OSA vs. 22 with RI Viral genome detection was significantly greater in the OSA group (17 vs. 0; *p* < 0.001). Rate of virus detection was significantly superior in the OSA group for rhinovirus (33.3%; *p* < 0.013), HAdV (29.2%; *p* > 0.024) and hMPV (25%; *p* < 0.041). OAHI was significantly enhanced in the OSA group (12.1 ± 4.08 vs. 0.61 ± 0.22 events/h sleep; *p* < 0.001) as well as significantly lower oxygen saturation (84.2 ± 5.36 vs. 94.2 ± 2.14%; *p* < 0.001).
Chen et al. [25]	Examination dossiers from Taiwan National Health Insurance Research Database (ICD-9-CM) from 2005 to December of 2015.	326 (182 in the EV infection group vs. 134 in the non-EV infection group) children were diagnosed with OSA during the study follow-up period. Hospitalization-associated EV infection was certainly associated with OSA after adjustment for age, sex, urbanization perinatal complications and atopic disease (aHR = 1.62, 95% CI: 1.18–2.21; *p* = 0.003).
Gutierrez et al. [23]	Primary care EMR to extract ICD-9/ICD-10 codes directly from the patients’ medical files in the BBC database from October 1998 to March 2019. Perform separate analyses using survey data collected independent of EMR during face-to-face follow-up visits to the BBC.	LRTIs occurring in early childhood (0 to 2 years) significantly increased the risk of pediatric OSA at the age of 5 years (*p* < 0.000). Severe preterm birth (GA ≤ 32 weeks) (*p* = 0.000), maternal obesity (BMI ≥ 30) (*p* = 0.001) and infant obesity (*p* = 0.000) exhibited significant associations with the development of OSA. Infants with severe RSV bronchiolitis present an increased risk of OSA within the first five years of life (OR 2.09, 95% CI 1.12 to 3.88, *p* = 0.020).

^1^ modified from Gozal [30] to assign numerical scores to each of the answers ranging from 0 (never), 1 (rarely), 2 (occasionally), 3 (frequently) to 4 (almost always). * Brodsky scores [27].

## Data Availability

The data underlying this article are available in the article.

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
