# Peer review of "Is There an Association between Viral Infections and Risk for Pediatric Obstructive Sleep Apnea? A Systematic Review"

_children, 2023, doi:10.3390/children10030487_

Round 1

Reviewer 1 Report

The authors have written a well designed and carefully prepared review of viral infections in children with obstructive sleep apnea. It is in accordance with the Preferred Reporting Items for Systematic Reviews and Meta-Analyses (PRISMA) 2020 guidelines and includes relevant and new literature. The review gives a very good overview of the currently available literature on this topic. Limitations of the review and the conclusions drawn from this are presented.

I have only a few comments:

The page numbers are not continuous.

In table 2 a more precise labeling of inclusion and exclusion criteria would facilitate reading.

In the last paragraph of the “Strength and limitations” section the sentence “The third limitation of the included studies was that, among the four included studies.” does not make sense.

Also as a limitation it should be more clearly stated that only one of the publications had a clear definition of how OSA was diagnosed.

In summery this well written review will give readers of “children” a good overview of relevant literature concerning the association of viral pathogens and pediatric OSA.

Author Response

We thank the reviewer for the helpful comments and suggestions.

Please provide your response for Point 1 : the page numbering has been changed.

Please provide your response for Point 2 : We have changed inclusion and exclusion criteria in table 2  in order to facilitate reading. 

Please provide your response for Point 3 : We have modified in the last paragraph of the “Strength and limitations” section the sentence “The third limitation of the included studies was that, among the four included studies.” => The third limitation of the included studies was that, among the four included studies, three presented a judgement serious of risk of bias [22–24] and one presented a moderate risk of bias [25]. 

Please provide your response for Point 4 : We added a sentence  in the last paragraph of the “Strength and limitations”. => In fact, only one study had a clear explanation of OSA diagnosis. 

Reviewer 2 Report

The authors conduct a major systematic review of the published literature on the cause-effect relationship of viral infections in childhood sleep apnoea syndrome.

This was the first review on the possible impact of a viral infection on OSA in childhood that used a systematic and objective approach to identify as many relevant literature as possible in English, hopefully to provide a comprehensive and balanced presentation of the topic about the role of respiratory viruses involving the development of AT hypertrophy and nasopharyngeal airway obstruction and pediatric OSA.

The authors follow an appropriate methodology using the PECO format, thoroughly analysing the selected articles. The authors express the limitations of the study. It is written in a fluent and understandable way. In my opinion, the work can be published in its current format.

Author Response

We thank the reviewer for his comments.

Reviewer 3 Report

I cannot think of any suggestions to improve this job.

Author Response

We thank the reviewer for his comments.